# Polymorphisms in the Genes Coding for TLRs, NLRs and RLRs Are Associated with Clinical Parameters of Patients with Acute Myeloid Leukemia

**DOI:** 10.3390/ijms23179593

**Published:** 2022-08-24

**Authors:** Katarzyna Wicherska-Pawłowska, Katarzyna Bogunia-Kubik, Bartłomiej Kuszczak, Piotr Łacina, Marta Dratwa, Bożena Jaźwiec, Tomasz Wróbel, Justyna Rybka

**Affiliations:** 1Department and Clinic of Hematology, Blood Neoplasms and Bone Marrow Transplantation of Wroclaw Medical University, 50-367 Wroclaw, Poland; 2Laboratory of Clinical Immunogenetics and Pharmacogenetics, Hirszfeld Institute of Immunology and Experimental Therapy, Polish Academy of Sciences, 53-114 Wroclaw, Poland

**Keywords:** single-nucleotide polymorphisms, innate immunity, acute myeloid leukemia

## Abstract

Toll-like receptors (TLRs), NOD-like receptors (NLRs), and RIG-I-like receptors (RLRs) are major elements of the innate immune system that recognize pathogen-associated molecular patterns. Single-nucleotide polymorphisms (SNPs) in the TLR, NLR, and RLR genes may lead to an imbalance in the production of pro- and anti-inflammatory cytokines, changes in susceptibility to infections, the development of diseases, and carcinogenesis. Acute myeloid leukemia (AML) is a bone marrow malignancy characterized by uncontrolled proliferation of transformed myeloid precursors. We retrospectively analyzed 90 AML patients. We investigated the effect of fifteen SNPs located in the genes coding for RLR1 (rs9695310, rs10738889, rs10813831), NOD1 (rs2075820, rs6958571), NOD2 (rs2066845, rs2066847, rs2066844), TLR3 (rs5743305, rs3775296, 3775291), TLR4 (rs4986791, rs4986790), and TLR9 (rs187084, rs5743836). We observed that *TLR4* rs4986791, *TLR9* rs5743836, and *NOD2* rs2066847 were associated with CRP levels, while *RLR-1* rs10738889 was associated with LDH level. Furthermore, we found *TLR3* rs5743305 *AA* to be more common in patients with infections. We also found *TLR9* rs187084 *C* to be associated with more favorable risk, and RLR-1 rs9695310 *GG* with higher age at diagnosis. In conclusion, the current study showed that SNPs in the genes encoding TLRs, NLRs, and RLRs may be potential biomarkers in patients with AML.

## 1. Introduction

Toll-like receptors (TLRs), NOD-like receptors (NLRs), and RIG-I-like receptors (RLRs) play essential roles in mechanisms of innate immunity. They detect pathogen-associated molecular patterns (PAMPs) from bacteria, viruses, fungi, protozoa, and other microorganisms that enter the human body. Detection of pathogen invasion activates intracellular transmission pathways, ultimately leading to increased production of proinflammatory cytokines and infected-cell death, as well as activation of adaptive immunity mechanisms [1,2,3]. In addition, TLRs, NLRs, and RLRs detect host cell damage-associated molecular patterns (DAMPs), which include free nucleic acids, uric acid crystals, heat shock proteins, and many others. Detection of DAMPs leads to the repair and regeneration of damaged tissues [4]. In addition to their involvement in immune mechanisms, these receptors are also involved in cell differentiation, maturation, apoptosis, and angiogenesis. The proper regulation of TLR, NLR, and RLR receptor activity and suppression of inflammatory response after infection is extremely important because chronic overproduction of proinflammatory cytokines is responsible for the development of chronic inflammatory, metabolic, neurodegenerative, and autoimmune diseases and cancers [5,6,7].

Single-nucleotide polymorphisms (SNPs) in genes encoding TLRs, NLRs, and RLRs can lead to an imbalance in the production of pro- and anti-inflammatory cytokines, changes in susceptibility to certain infections, and the development of allergic and inflammatory diseases, as well as carcinogenesis [8,9]. In addition, some SNPs are associated with increased resistance of cancer cells to treatment and apoptosis [10,11]. The search for associations between the occurrence of specific SNPs and the predisposition to develop various cancers and their impact on disease course and prognosis has been ongoing for several years.

Acute myeloid leukemia (AML) belongs to the group of malignant blood cancers and is the most common acute leukemia among adult patients. It leads to the suppression of normal hematopoiesis in the bone marrow and, therefore, to an increased risk of severe infections, anemia, and thrombocytopenia. AML is heterogeneous and differs in its course, degree of resistance to treatment, ability of blasts to cross the blood–marrow barrier, and formation of extramedullary metastases. Prognosis depends on the presence of specific molecular and cytogenetic mutations in tumor cells, and 5-year patient survival rates range from 20–30% [12]. Infections associated with neutropenia after chemotherapy are a major problem during AML therapy. It is not uncommon for patients to develop sepsis and septic shock that leads to death. Therefore, it is important to identify patients who are at a high risk of severe infections and to apply infection prevention and treatment measures. In order to achieve better AML treatment outcomes, an individualized approach to the patient is necessary, taking into account factors that may affect the course of therapy.

The aim of this study was to present 15 SNPs in TLRs, NLRs, and RLRs encoding genes in patients with de novo diagnosed AML and their association with clinical features such as infection rate, blood CRP level, cytogenetic risk according to the European Leukemia Net (ELN), and tumor burden (number of blasts in the bone marrow and presence of extramedullary metastases, blood LDH levels) at diagnosis, as well as their relationship with patients’ age and sex. Due to heterogeneity of the course, response to treatment, prognosis, and the survival of young and elderly patients diagnosed with AML, at a later stage of the study, we analyzed the obtained results of two age groups: >55 years of age and ≤55 years of age.

## 2. Results

### 2.1. SNPs in Toll-like Receptors Genes

We found that AML patients with favorable or intermediate risk at diagnosis (according to ELN) were more likely to have allele rs187084 *C* in the *TLR9* gene than patients with unfavorable risk (*p* = 0.012), Figure 1. Patients with rs4986791 *T* in the *TLR4* gene had lower CRP levels than CC homozygotes (*p* = 0.015), with rs4986791 *T* in the *TLR4* gene being more common in those with CRP levels < 5 mg/L (*p* = 0.028), Figure 2. Another polymorphism associated with lower CRP levels was rs5743836 *C* in the *TLR9* gene. Carriers of this SNP had lower serum CRP than rs5743836 *TT* homozygotes (*p* = 0.048), Figure 3. Additionally, patients with rs3775291 *G* in the *TLR3* gene tended to have higher LDH levels at diagnosis than rs3775291 *AA* homozygotes (*p* = 0.053). Infectious complications (all types of bacterial, viral, and fungal infections except for SARS-CoV-2 infections) during AML therapy were observed more frequently in *TLR3* rs5743305 *AA* homozygotes (*p* = 0.015), Figure 4. The overall survival (OS) analysis showed no statistically significant differences in OS among patients with *TLR3* rs5743305 *AA* genotype and the *TLR3* rs5743305 *AT* and *TT* genotypes—in the group with infections during AML treatment or in the group without infections (*p*-value, 0.711 and 0.465, respectively), Figure 5 and Figure 6. Extramedullary metastases of AML tended to be more common in patients who carry rs3775296 *T* in the *TLR3* gene and rs4986790 *G* in the *TLR4* gene (*p* = 0.054 and *p* = 0.078, respectively). Patients who were diagnosed with disease metastases outside the bone marrow at the time of diagnosis tended to be less likely than patients with disease localized to the bone marrow to carry rs3775291 *A* in the *TLR3* gene and rs187084 *C* in the *TLR9* gene (*p* = 0.068 and *p* = 0.053, respectively). A summary of the correlations between different SNPs in genes coding for TLR3, TLR4, and TLR9 and clinical features in patients with AML is presented in Table 1. For statistically significant results, bold font is used.

### 2.2. SNPs in NOD-like Receptors Genes

We observed a statistically significant difference in patients with the *NOD2* rs2066847 ins variant (*ins/del, ins/ins*), who showed lower CRP protein levels at diagnosis than patients with rs2066847 *del/del* (*p* = 0.045), Figure 7. There was also a trend for higher LDH levels in patients with *NOD2* rs2066844 *T* compared to patients without the T allele (*p* = 0.062).

### 2.3. SNPs in the RIG-I-like Receptors Genes

*DDX58* is a gene encoding the DeXD/H-Box Helicase 58 or RIG-I receptor. Statistically significant associations were observed between different SNPs and clinically relevant features of AML. We observed that the rs10738889 *G* allele was associated with lower LDH levels than rs10738889 *AA* (*p* = 0.027, Figure 8). There was also a trend showing a difference in LDH levels in patients with rs10813831 *G* compared to patients with the *AA* genotype (*p* = 0.056). Regarding CRP levels, we observed that *DDX58* rs10813831 *A* tended to be less common in patients with CRP < 5 than in those with CRP > 5 (*p* = 0.098). The rs10738889 *A* allele was more common in male than in female AML patients (*p* = 0.033; Figure 9), and the rs9695310 *C* variant was associated with lower age at diagnosis than *GG*, Figure 10. Furthermore, while not statistically significant, a difference in the number of blast cells in the bone marrow at the diagnosis of AML was observed. A higher percentage of blast cells was present in patients with the rs10813831 *G* allele than rs10813831 *AA* (*p* = 0.065), while a lower percentage of blast cells was present in patients with rs10738889 *G* than rs10738889 *AA* (*p* = 0.082).

### 2.4. SNPs in TLRs, NLRs, and RLRs in Groups of Patients >55 yo and ≤55 yo

After dividing the study group into a group of younger (≤55 years old) and older (>55 years old) patients, the statistical analysis showed the following associations: *DDX58* rs10813831 *A* was more common among patients with infections than without infections in the younger (*p* = 0.046) but not in the elderly (*p* = 0.566) group. *TLR4* rs4986791 *T* was associated with a higher LDH level than CC (*p* = 0.044) among young but not in elderly patients (*p* = 0.171). The presence of extramedullary metastases among young patients was associated with a higher prevalence of genotypes *TLR3* rs3775296 *T* (*p* = 0.036) and *TLR4* rs4986790 *G* (*p* = 0.020). These correlations did not occur in the elderly group (*p* = 0.593 and *p* = 0.593, respectively). Additionally, *TLR9* rs187084 *C* was more common in patients without extramedullary lesions in younger patients (*p* = 0.035), but not in older patients (*p* = 0.623). In the group of elderly patients, we showed statistically significant associations: *TLR9* rs187084 *C* was more common among patients with favorable and intermediate risk according to ELN (*p* = 0.005); *TLR4* rs4986791 *T* and *TLR9* rs5743836 *C* were associated with lower CRP level at diagnosis of AML (*p* = 0.009 and *p* = 0.003, respectively). In the group of younger patients, the above associations were not statistically significant (*p* = 0.481, *p* = 0.204, and *p* = 0.523, respectively).

## 3. Discussion

The results of our study presented here are among the first to describe so extensively the associations between SNPs in genes encoding pattern recognition receptors belonging to the innate immune system and clinical features of patients with AML. Currently, little is known about the impact of SNPs in genes encoding TLRs, NLRs, and RLRs on the course and prognosis of hematologic malignancies. M. Quirino et al. [13] studied the effect of polymorphisms in TLRs genes on the development of myeloproliferative neoplasms (MPNs). In that study, they found that *TLR9*-1486 *CT* may be associated with a lower susceptibility to developing polycythemia vera (PV), and in the haplotype frequency analysis, *TLR9*-1237*T*/-1486*C* was less common in men compared to controls, as well as in men negative for JAK V617F. Y. Zhou et al. [14] showed that inflammasome-related genes (NLRP3, NF-κB1, CARD8, IL-1β, and IL-18) were highly expressed in patients with MPNs and that the NF-κB1 polymorphism (rs28362491) was more common in MPN patients than in the study group. The effects of SNPs on the NLRP3 inflammasome system have also been studied as risk factors for the development and severity of AML. In a study by H. Wang et al. [15], *IL-1B* (rs16944) *GA* was more prevalent in patients with cytogenetic favorable risk, while the number of bone marrow blasts in patients with *IL-18* (rs1946518) *GG* or *GT* genotypes was higher than in *TT* patients. Furthermore, the *IL-18* rs1946518 *GT* genotype was statistically significantly associated with worse AML overall survival.

Patients diagnosed with AML and undergoing chemotherapy are a group of patients at particularly high risk of developing bacterial and fungal infections. Approximately 80% of patients develop neutropenic fever (NF) during AML therapy, with the etiologic agent being detected in less than 50% of cases [16]. The most commonly observed infectious complications are pneumonia, gastrointestinal infections, urinary tract infections, and central vascular catheter-related infections [17]. Invasive fungal infections are a major problem when treating patients with AML. Their occurrence is favored by prolonged neutropenia, older patient age, and low albumin levels [18]. The treatment of fungal infections in the era of new anticancer drugs poses many difficulties due to the interaction of most new molecules with antifungal drugs [16]. Mechanisms of innate immunity are the body’s first line of defense against microbial invasion; their efficient functioning provides protection against the development of severe infections. In the present study, a significant statistical association between the *TLR3* rs5743305 *AA* genotype and an increased frequency of infectious complications (non-SARS-CoV-2 infections) was detected. The TLR3 receptor is responsible for recognizing viral dsRNA in cell endosomes. TLR3 activation leads to the production of increased amounts of interferon type I and the death of infected cells [19]. *TLR3* rs5743305 homozygosity may cause malfunction and make cells much more susceptible to viral infections. Antiviral prophylaxis should be implemented in this group of patients, and they should be closely monitored for the development of viral infections that can lead to the patient’s death.

Age at diagnosis is a clinically relevant parameter that indirectly influences the treatment plan of AML patients. In younger patients, intensive chemotherapy protocols are generally possible, whereas in older patients (>65–70 years of age), less intensive regimens are applicable [20]. In a study [21] of 13,283 patients diagnosed with AML, Acharya et al. showed that older patient age and male gender were associated with worse 3-year overall survival in univariate analysis, and these parameters remained independent prognostic factors in multivariate analyses. In our study group, the mean age was 56 years. Patients with the *DDX58* rs9695310 *CC* genotype had the lowest age at diagnosis (51 years), *GC* heterozygotes 55 years, and *GG* homozygotes 64 years. Polymorphisms in the *NOD1* and *NOD2* genes (rs6958571 *AA* and rs2066847 *ins*, respectively) also showed an association with lower patient age at AML diagnosis, but this value was not statistically significant. Regarding gender, we observed that the *DDX58* rs10738889 *A* allele was more frequently present among male than female patients with AML.

Moreover, we found that the *TLR9* rs187084 *C* allele was less frequent among patients with unfavorable cytogenetic and molecular prognosis than in other risk groups. Patients with unfavorable risk according to ELN are a group that requires particularly intensive treatment, as a resistance to chemotherapy and rapid relapse are more common [20,21].

The number of blasts in the bone marrow at diagnosis of AML does not show a significant impact on disease course, treatment, and prognosis. Nevertheless, we observed an association between individual polymorphisms in the DDX58 gene and a higher percentage of blasts in the bone marrow at diagnosis. Patients with the *DDX58* rs10813831 *G* allele and *DDX58* rs10738889 *AA* genotypes tended to have a higher percentage of tumor cells in the bone marrow than carriers of *DDX58* rs10813831 *AA* and *DDX58* rs10738889 *G*. The DDX58 gene encodes the intracellular receptor RIG-I, which is responsible for detecting foreign double-stranded RNA (dsRNA). Activation of RIG-I leads to the increased production of proinflammatory cytokines (mainly interferon) as well as inflammasome formation and the induction of pyroptosis-induced death of infected cells [22,23]. The aforementioned SNPs in the DDX58 gene may be associated with a predisposition to increased and prolonged production of proinflammatory cytokines, which stimulate bone marrow stem cells to mutate into cancer cells and create the right conditions in the bone marrow microenvironment for mutant cells to proliferate [24,25].

In our present study, AML extramedullary involvement (EMI) metastases were found in 13 patients (14% of all patients). EMI metastases are most commonly found in the skin, central nervous system, lymph nodes, and less commonly, in internal organs and bones. L. Fianchi et al. [26] described factors that increase the risk of extramedullary metastases in AML. EMI was more common in patients with monocytic and myelomonocytic AML subtypes, and blasts were characterized by a lack of CD117 on immunophenotyping; molecularly, MLL gene (mixed-lineage leukemia gene) rearrangement was most commonly diagnosed, and cytogenetically, trisomy of chromosome 8 was most commonly detected. The same study confirmed an unfavorable prognosis in patients with extramedullary metastases with a median survival of 11.6 months. The factors that improve the survival of patients with EMI are intensive chemotherapy treatment, achieving complete remission, and performing an alloHSCT procedure. Our analysis found an association between SNPs in genes encoding TLR3, TLR4, and TLR9 receptors and the frequency of extramedullary metastases. The *TLR3* rs3775296 *T* allele and the *TLR4* rs4986790 *G* allele were more frequent in patients with EMI than in those without extramedullary metastases, whereas the *TLR3* 3775291 *A* allele and the *TLR9* rs187084 *C* allele tended to be less frequent in patients with EMI than in those without extramedullary manifestations of AML. Statistical significance was not demonstrated for all determinations. The study group size would need to be increased to confirm these preliminary correlations.

CRP is an acute-phase protein whose serum levels correlate closely with the severity of the inflammatory response. Inflammation has been a known carcinogen for many years. It contributes to tumor growth, cancer cell invasion, migration, and metastasis [27]. Shrotriya et al. [28] analyzed 271 articles on the correlation of CRP levels with prognosis in patients with solid tumors. High CRP levels were associated with increased mortality in 90% of cases, especially among patients with gastrointestinal and renal cancers. In addition, high CRP levels were associated with poor response to treatment and increased relapse rates. We found much less data on the association of CRP levels with prognosis for patients with AML. Gradel et al. [29] found that high CRP levels at the time of AML diagnosis are associated with poorer patient general status. Patients in WHO 3/4 general status had 68 mg/L higher CRP levels than patients in WHO 0 status. We observed associations between the *TLR4* rs4986791 *T* allele and the *TLR9* rs5743836 *C* allele and lower CRP levels compared to *TLR4* rs4986791 *CC* and *TLR9* rs5743836 *TT*. Among NOD-like receptors, a statistically significant correlation was described for *NOD2* rs2066847 *ins*, which was associated with lower CRP than *NOD2* rs2066847 *del/del*. The *DDX58* rs10813831 *A* allele tended to be more frequent in patients with CRP > 5 mg/dL at AML diagnosis, although this was not statistically significant.

Lactate dehydrogenase (LDH) is an enzyme released from ruptured cells. In aggressive diseases with rapid cell turnover, blood LDH levels increase significantly. High LDH levels at diagnosis have been described as a factor for poor prognosis in patients with AML [30,31], correlated with shorter 1-year overall survival and increased 60-day mortality compared to patients with slightly elevated blood LDH levels. In the present study, we found that the *DDX58* rs10738889 *G* allele was associated with lower LDH levels than rs1078889 *AA.* Similarly, *DDX58* rs10738889 *GG* was associated with lower LDH levels than rs10738889 *AG*/*AA*. Furthermore, we observed a trend of higher LDH levels in patients with the *DDX58* rs10813831 *G* allele, *NOD2* rs2066844 *T* allele, and *TLR3* 3775291 *G* allele. Patients with high LDH levels should be closely monitored and may require more intensive treatment protocols.

## 4. Materials and Methods

### 4.1. Subjects

In our study, we retrospectively analyzed 90 patients diagnosed with AML treated at the Department of Hematology, Blood Neoplasms and Bone Marrow Transplantation of Wroclaw Medical University between 2018 and 2020. There were 42 men and 48 women in the study group. The median age was 61 years (range 21–81 years), and the number of patients aged over 55 years was 54. All patients suffered from primary AML. The characteristics of the study group can be found in Table 2. The research was approved by the Bioethics Committee at the Wroclaw Medical University. All participants signed informed consent forms to participate in the study.

### 4.2. DNA Isolation, Genotyping

Genomic DNA was isolated from peripheral blood taken on EDTA from 90 AML patients using the commercial GeneMATRIX Quick Blood DNA Purification kit from EurX (Gdańsk, Poland) and NucleoSpin Blood kits (MACHEREY-NAGEL GmbH & Co. KG, Dueren, Germany) according to the manufacturers’ protocols. DNA concentration and purity were assessed on the DeNovix DS-11 spectrophotometer (DeNovix Inc., Wilmington, DE, USA). Isolated DNA was stored at −20 °C for further use.

Patients were genotyped for the DDX58 (rs9695310, rs10738889, rs10813831), NOD1 (rs2075820, rs6958571), NOD2 (rs2066845, rs2066847, rs2066844), TLR3 (rs5743305, rs3775296, 3775291), TLR4 (rs4986791, rs4986790), and TLR9 (rs187084, rs5743836) genetic variants using LightSNiP assays (TIB MOLBIOL, Berlin, Germany), and real-time PCR was performed on a LightCycler 480 II device (Roche Diagnostics, Rotkreuz, Switzerland) according to the manufacturers’ instructions.

### 4.3. Statistical Analysis

Statistical analysis of the results obtained was performed using the Real Statistics Resource Pack for Microsoft Excel 2013 (version 15.0.5023.1000, Microsoft, Redmont, WA, USA). Mann–Whitney U test was used to determine the associations between individual SNPs and CRP level, LDH level, and blast cell count in the bone marrow, while Fisher’s exact test was used to determine the associations between different SNPs and age of onset, presence of extramedullary metastases, presence of infection during treatment, and to illustrate the associations of SNPs with risk groups according to ELN. Kaplan–Meier curves and the Gehan-Breslow-Wilcoxon test were used for overall survival analysis. *p*-values < 0.05 were regarded as statistically significant. Graphs showing the obtained results were created using GraphPad Prism (GraphPad Software, La Jolla, CA, USA, version 8.0.1).

## 5. Conclusions

In conclusion, we demonstrated correlations between various SNPs in genes encoding Toll-like, NOD-like, and RIG-I-like receptors and specific clinical features of AML patients that may affect prognosis. These polymorphisms may be used as prognostic markers in AML in the future, but their potential implementation into clinical practice requires further studies on larger groups of patients.

## Figures and Tables

**Figure 1 ijms-23-09593-f001:**
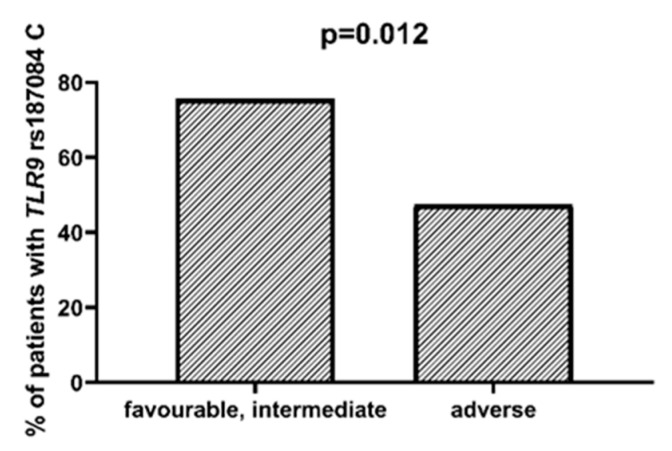
Association between *TLR9* rs187084 *C* and risk according to ELN.

**Figure 2 ijms-23-09593-f002:**
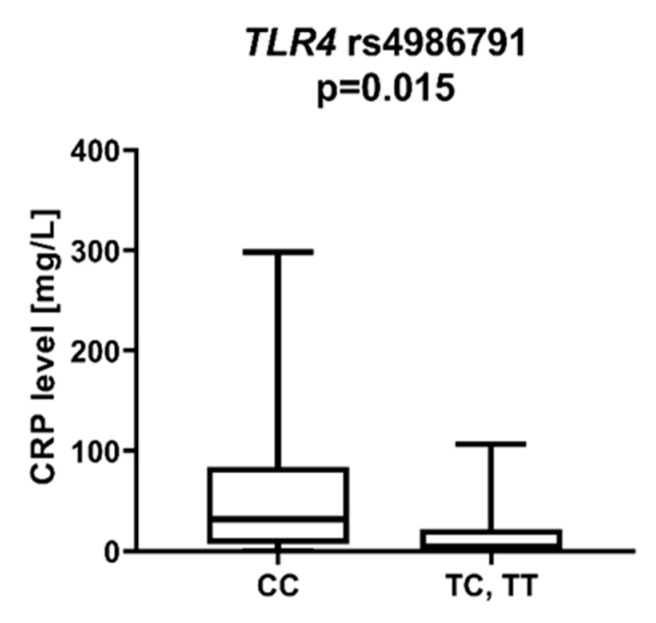
CRP level in patients with and without *TLR4* rs4986791 *T.*

**Figure 3 ijms-23-09593-f003:**
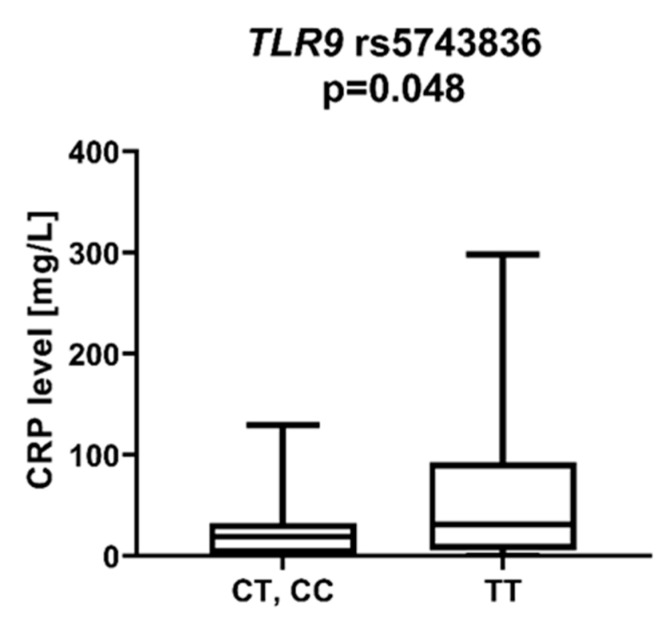
CRP level in patients with and without *TLR9* rs5743836 *C*.

**Figure 4 ijms-23-09593-f004:**
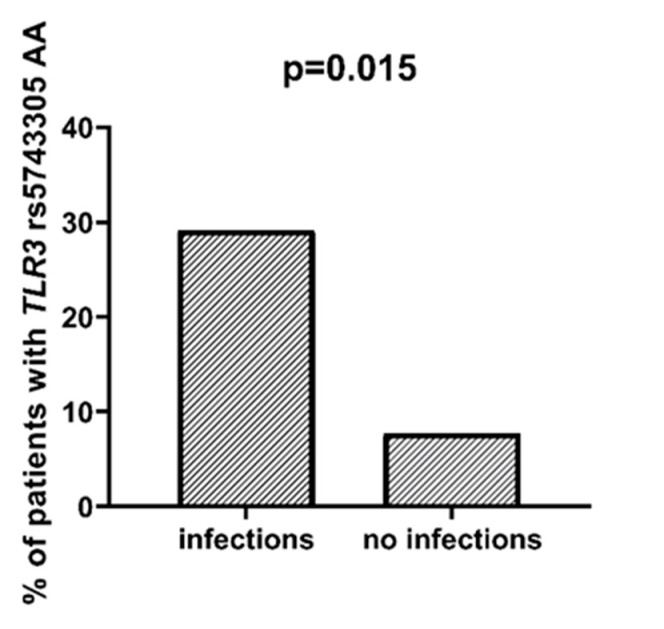
Association between *TLR3* rs5743305 *AA* and incidence of infection.

**Figure 5 ijms-23-09593-f005:**
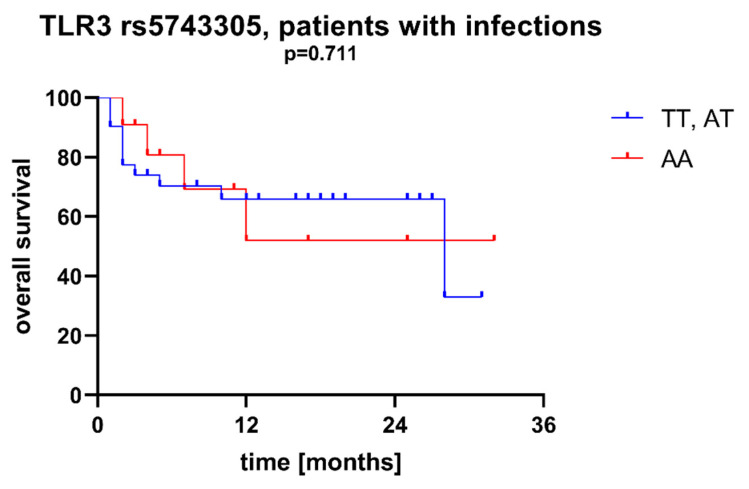
Overall survival of patients with different SNPs in *TLR3* rs5743305—patients with infections during AML therapy.

**Figure 6 ijms-23-09593-f006:**
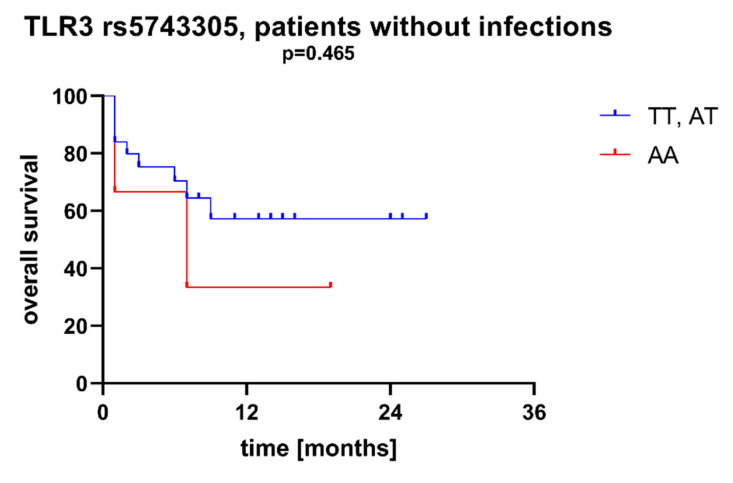
Overall survival of patients with different SNPs in *TLR3* rs5743305—patients without infections during AML therapy.

**Figure 7 ijms-23-09593-f007:**
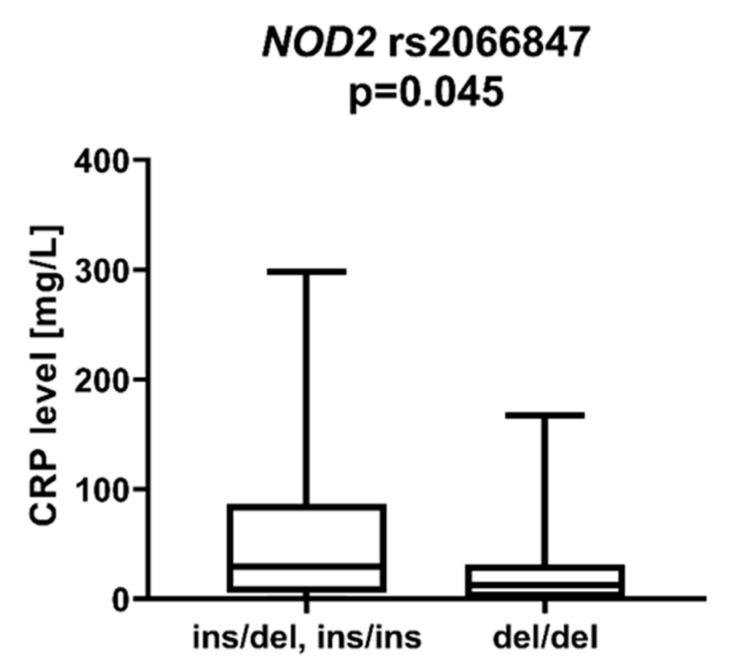
CRP level in patients with and without rs2066847 insertion in *NOD2* gene.

**Figure 8 ijms-23-09593-f008:**
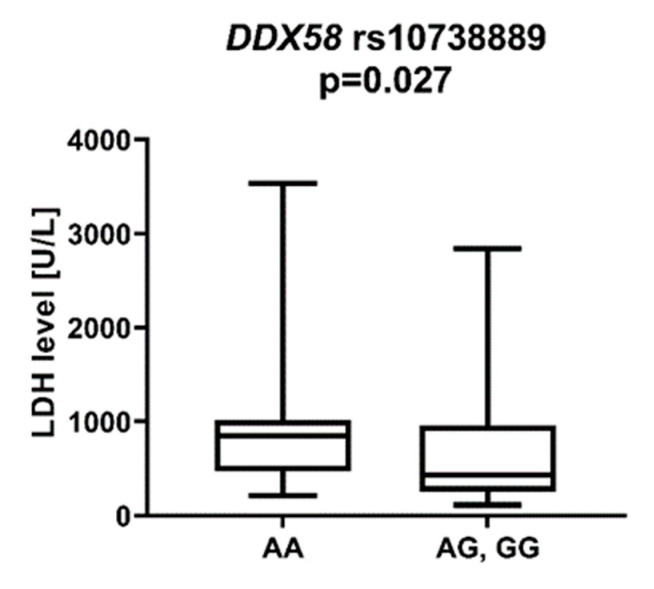
LDH level in patients with *DDX58* rs10738889 *G* allele.

**Figure 9 ijms-23-09593-f009:**
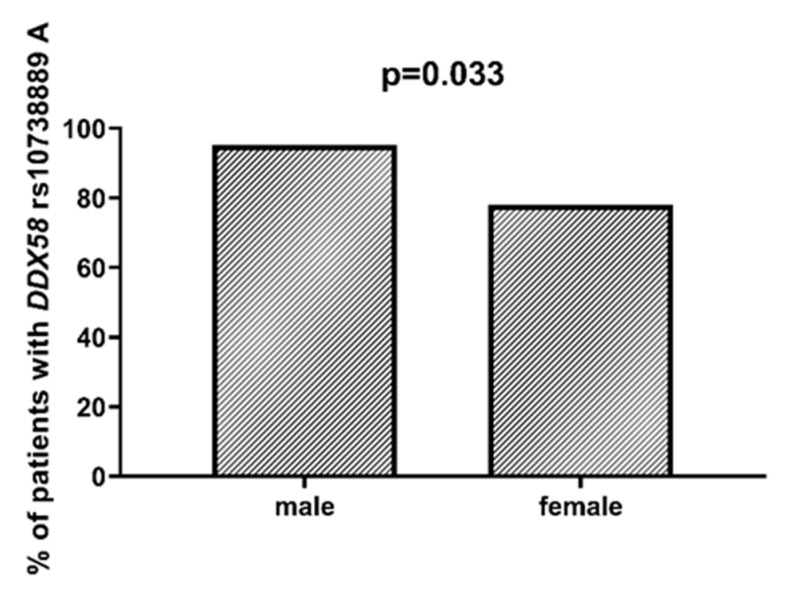
Association between *DDX58* rs10738889 *A* and gender.

**Figure 10 ijms-23-09593-f010:**
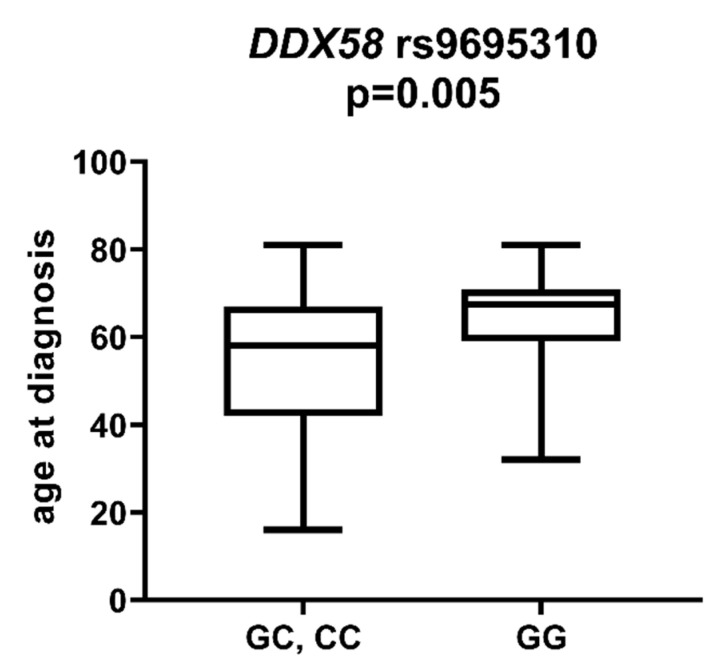
Age at diagnosis in patients with and without *DDX58* rs9695310 *C* allele.

**Table 1 ijms-23-09593-t001:** Summary of correlations between SNPs in genes coding for TLR3, TLR4, and TLR9 and different clinical features in patients with AML. For statistically significant results (*p* < 0.05), bold font is used.

	SNPs in Gene for TLR3	SNPs in Gene for TLR4	SNPs in Gene for TLR9
Favorable/intermediate risk according to ELN			**rs187084 C**
Lower CRP level		**rs4986791 T**	**rs5743836 C**
Higher CRP level		**rs4986791 CC**	**rs5743836 TT**
Lower LDH level	rs3775291 A		
Higher LDH level	rs3775291 G		
Higher risk of infection	**rs5743305 AA**		
Extramedullary metastases	rs3775296 T	rs4986790 G	
No extramedullary metastases	rs3775291 A		rs187084 C

**Table 2 ijms-23-09593-t002:** Study group characteristics. LDH—lactate dehydrogenase, ELN—European Leukemia Net.

Characteristic	Study Group (n = 90)
Median age	61 (range 21–81)
Age ≤ 55 yo	36
Age > 55 yo	54
Gender	
Female	48
Male	42
% of blasts in bone marrow at diagnosis	
On average	56
<56	41
≥56	48
LDH (U/L) at diagnosis	
Median	540
Within normal range (<220 U/L)	8
Elevated (≥220 U/L)	82
CRP (mg/L) at diagnosis	
Median	31.15
Within normal range (<5 mg/L)	21
Elevated (≥5 mg/L)	69
Extramedullary metastases at diagnosis	
Yes	13
No	77
Risk according to ELN	
Favorable/intermediate	52
Unfavorable	38
Infectious complications during treatment	
Yes	50
No	40

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
