# Peer review of "Polymorphisms in the Genes Coding for TLRs, NLRs and RLRs Are Associated with Clinical Parameters of Patients with Acute Myeloid Leukemia"

_ijms, 2022, doi:10.3390/ijms23179593_

Round 1

Reviewer 1 Report

The authors have examined the potential impact of a series of SNPs associated with Toll-like receptors (TLRs), NOD-like receptors (NLRs) and RIG-I-like receptors (RLRs) in Acute myeloid leukemia.  

It is by nature of examining SNPs that the numbers can be very repetitive and difficult for the reader to follow.  This is particularly evident in Section 2.1 where the clinical parameter is associated with each SNP; I wonder if might be more relevant for each SNP to associated with the clinical parameters or perhaps a summary table would be useful.   

Also many of the correlations are greater than 0.05 - although they do say tended to be associated - but without this virtually all correlations were not statistically significant.  

Figures 1 - 3 could be combined into one figure.   The increased rate of infections associated with some SNPs is interesting and have the authors considered looking at overall survival or risk group status for these patients.   Are they for example primary or secondary AML patients?

Overall, some interesting data that could be expanded in to a more clinically relevant study.

Reviewer 2 Report

Considering the high heterogeneity of AML between young and elderly (biology, responses, infections, outcomes, etc) it is relevant to distinguish the analysis according to the age (material methods, results and conclusion)

Round 2

Reviewer 1 Report

The authors have addressed my previous comments